# It's All Relative: Relative Uncertainty in Latent Spaces using Relative Representations

**Fabian Martin Mager**
Technical University of Denmark
fmager@dtu.dk

**Valentino Maiorca**
Sapienza University of Rome, Italy
maiorca@di.uniroma1.it

**Lars Kai Hansen**
Technical University of Denmark
lkai@dtu.dk

**Editors:** Marco Fumero, Clementine Domine, Zorah Lähner, Donato Crisostomi, Luca Moschella, Kimberly Stachenfeld

## Abstract

Many explainable artificial intelligence (XAI) methods investigate the embedding space of a given neural network. Uncertainty quantification in these spaces can lead to a better understanding of the mechanisms learned by the given network. When concerned with the uncertainty of functions in latent spaces we can invoke ensembles of trained models. Such ensembles can be confounded by reparameterization, i.e., lack of identifiability. We consider two mechanisms for reducing reparametrization "noise", one based on relative representations and one based on interpolation in weight space. By sampling embedding spaces along a curve connecting two fully converged networks without an increase in loss, we show that the latent uncertainty becomes overestimated when comparing embedding spaces without considering the reparametrization issue. By changing the absolute embedding space to a space of relative proximity, we show that the spaces become aligned, and the measured uncertainty decreases. Using this method, we show that the most non-trivial changes to the latent space occur around the midpoint of the curve connecting the independently trained networks.

## 1 Introduction

A neural network (NN) trained to perform a certain downstream task, e.g., regression or classification, learns a mapping $f : \mathbf{x}_i \rightarrow \mathbf{y}_i$ given a set of observations $\mathcal{D} = \{(\mathbf{x}_i, \mathbf{y}_i)\}_i^N$, where $\mathbf{x}_i = [x_1, x_2, ..., x_D]$ and $N$ is the number of available observations and $D$ is the latent space dimension. The learned function $f$ is hierarchical, i.e. $f(\mathbf{x}) = h_L(h_{L-1}(...h_1(\mathbf{x})))$ where the intermediate representation of the observation $\mathbf{x}_i$ after layer $l$ is denoted as $\mathbf{z}_i$. We note the mapping of the observation $\mathbf{x}_i$ to its latent representation $\mathbf{z}_i$ as $h_l : \mathbf{x}_i \rightarrow \mathbf{z}_i$. The latent space is assumed to be semantically meaningful for relevant downstream tasks. Therefore, many XAI methods investigate the properties of the latent space of a NN, for example, its geometry [9] or the semantic structure [5]. XAI has many dimensions [11], one of which is referred to as *local* vs. *global*. Local explainability concerns predictions of individual samples, whereas global explainability considers the network as a whole. Most local methods give attributions to the input features $[x_1, x_2, ..., x_D]$, which are important to the output $\hat{\mathbf{y}}$. Recently, Wickstrøm et al. [10] proposed a method that maps attributions to input features based on the importance of its latent vector $\mathbf{z}$. In contrast, global explainability methods such

as *Concept Activation Vectors* (TCAV) [5] aim to explain the entire latent space based on semantics. One persistent challenge in assessing geometry and semantic structure of the latent space is the *reparametrization issue*, driven by the fact that there is no "set of optimal parameters and that we can always parametrize the manifold in a different, but equally good, way" ([2]). In other words, while the end-to-end function $f$ remains the same, different parametrizations lead to different but equally performing embedding functions $h$.

Moschella et al. [7] introduced Relative Representations, which define an alternative representation of the latent space. They empirically show that the latent spaces of two identical models trained on the same data but different initializations are identical up to angle-preserving transformations. Their proposed method uses a set of latent vectors $\mathcal{A} = \{\mathbf{z}_i\}_i^A$, called *anchors*, and redefines the position of each latent vector to be *relative* to the set of anchors, according to a similarity function $sim : \mathbb{R}^D \times \mathbb{R}^D \to \mathbb{R}$. Depending on the choice of the similarity function, the latent space becomes invariant to certain transformations.

The reparametrization issue is also related to the loss landscape in NNs. Several works have exploited the reparametrization issue to efficiently train ensembles of models, which outperform their single-model counterparts [4]. Garipov et al. [1] investigated the geometrical properties of loss landscapes and showed that two identical NNs trained on different seeds can be connected by a simple curve in weight space, such that the loss under the curve is constant. This curve $\phi$ parameterized by $\theta$ connects two points $\hat{w}_1$ and $\hat{w}_2$ in parameter space and is found by minimizing the loss $l(\theta) = \int_0^1 \mathcal{L}(\phi(t)), dt = \mathbb{E}_{t \sim U(0,1)}[\mathcal{L}(\phi_\theta(t))]]$, where $\mathcal{L}$ is the loss function used to find $\hat{w}_1$ and $\hat{w}_2$. We refer to section B for details on the curve-finding procedure. Once $\theta$ is fitted, one can sample from $\phi_\theta(t))$ for $0 \leq t \leq 1$ and build an ensemble of models from living on the curve $\phi_\theta(t)$.

Ensembling methods reduce both the bias and variance of $\hat{\mathbf{y}}$. The uncertainty over $K$ samples of the end-to-end function $f$ in the prediction of $\hat{y}_i$ is given by

$$\sigma^2(\hat{\mathbf{y}}_i) = \text{trace}\left(\mathbb{E}_{k \in K}[(f_k(\mathbf{x}_i) - \bar{\mathbf{y}}_i)(f_k(\mathbf{x}_i) - \bar{\mathbf{y}}_i)^T]\right) \tag{1}$$

where $\bar{\mathbf{y}}_i = \mathbb{E}_{k \in K}[f_k(\mathbf{x}_i)]$. Following the above reasoning, the embedding function's uncertainty $h$ could be obtained by replacing $f_k$ with $h_k$ and $\mathbf{y}_i$ with $\mathbf{z}_i$. Due to reparametrization however, the latent uncertainty $\sigma^2(\mathbf{z}_i)$ will be overestimated (see Appx. A). We argue that there are two factors contributing to the overall observed uncertainty $\sigma_\mathcal{O}^2(\mathbf{z}_i)$,

$$\sigma_\mathcal{O}^2(\mathbf{z}_i) = \sigma_\mathcal{R}^2(\mathbf{z}_i) + \sigma_\mathcal{M}^2(\mathbf{z}_i) , \tag{2}$$

where $\sigma_\mathcal{R}^2$ refers to the uncertainty caused by reparametrization and $\sigma_\mathcal{M}^2$ to the uncertainty of the model. In equation 2 we assume independence of the model and reparametrization uncertainty. Such independence could arise as a consequence of initialization being independent of the data. In order to isolate the model uncertainty $\sigma_\mathcal{M}^2(\mathbf{z}_i)$, the reparametrization uncertainty $\sigma_\mathcal{R}^2(\mathbf{z}_i) \to 0$. In Appx. A we show that, for rotation and scaling transformations, the Relative Representation framework, using a cosine similarity function, can eliminate $\sigma_\mathcal{R}^2(\mathbf{z}_i)$. In this work, we

- Define an alignment score $\rho$, which estimates the signal-to-noise ratio for a given ensemble of latent observations. The alignment score $\rho$ is a measure of separability between $N$ latent points given $K$ samples each, similar to Fisher's Linear Discriminant objective for $N$ classes.
- Show empirically that when sampling models along the curve $\phi$, transforming the embeddings into a space of relative proximity increases alignment between latent observations.
- Show that the latent observations for the curve $\phi(t)$ have high alignment around the endpoints and little alignment at the bending point, suggesting limited information gain when sampling around the endpoints and high information gain when sampling across the midpoint.

## 2 Methods

The intuition behind our alignment metric is the following: Given $K$ samples of a latent vector $\mathbf{z}_i$, where each sample is an embedding of a given observation $\mathbf{x}_i$ obtained from a unique function $h_k$, the alignment score $\rho_i$ should be high if all samples form a compact region in the latent space and are well separated from all other points (see Figure 1). The alignment score $\rho$ of the entire embedding space can be estimated as the average of individual scores. This formulation is similar to Fisher's Linear Discriminant (FLD) objective, which measures the separability of two or more classes of

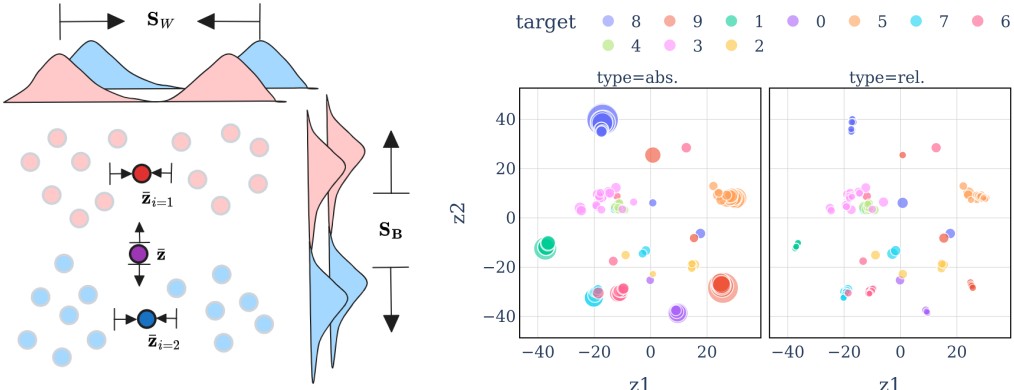

Figure 1: *Left*: Conceptual visualization of the alignment measure. Given the two observations $\mathbf{x}_{i=1}$ and $\mathbf{x}_{i=2}$ and $K$ realizations of their embeddings $\mathbf{z}_{i=1}$ and $\mathbf{z}_{i=2}$ obtained from a unique embedding function $h_k$, we seek small within-observation variance $\mathbf{S}_W$ relative to the between-observation variance $\mathbf{S}_B$. *Right*: Visualization of uncertainties in latent space using the ratio of within vs. between sample variance. The plots show two identical t-SNE projections of the VGG-16 model and 10 classes of CIFAR100. The dot size represents uncertainty measured in absolute space (left) and relative space (right) using a cosine similarity projection.

objects. The FLD objective is given by the ratio of between-class vs. within-class covariance. Given $K$ realizations of latent vector $\mathbf{z}_i$, the objective can be applied on the scale of a single observation, i.e. the ratio of between-observation vs. within-observation covariance. Due to the high dimensionality of $\mathbf{z}$ and the few number of available samples, it is not feasible to calculate the full covariance matrix. We therefore use the variance estimator and express the ratio as a signal-to-noise ratio, such that the alignment metric $\rho$ is bounded $0 \leq \rho \leq 1$. We define the within-observation variance $\sigma_W^2$ and between-observation variance $\sigma_B^2$ of $k$ realizations of the latent space and their alignment $\rho$ as

$$\sigma_W^2 = \frac{1}{N} \sum_{i=1}^{N} \sum_{k=1}^{K} (\mathbf{z}_{ik} - \bar{\mathbf{z}}_i)^2 \tag{3}$$

$$\sigma_B^2 = \frac{1}{N} \sum_{i=1}^{N} (\bar{\mathbf{z}}_i - \mathbb{E}(\bar{\mathbf{z}}_i))^2 \tag{4}$$

$$\rho = \frac{\sigma_B^2}{\sigma_B^2 + \sigma_W^2} \tag{5}$$

We conduct the following experiment independently using a VGG-16 [8] and a Preactivation-ResNet [3] on CIFAR-100 [6]. First, fully train on three different seeds until convergence, resulting in three unique models $\hat{w}_1$, $\hat{w}_2$, and $\hat{w}_3$. For each pairwise combination of seeds, we fit a Bezier curve with fixed endpoints and a single bend following the procedure described in [1]. Once a curve $\phi_{ij}(t)$ between two modes $\hat{w}_i$, $\hat{w}_j$ is found, we sample $K$ models at steps $t = k * \Delta t$, where $k \in [0, ..., K-1]$ and $\Delta t = \frac{1}{K}$. For all experiments, we set $K = 21$. We proceed by evaluating the performance at each step $t$, as well as the ensemble performance over all models up to step $t$ using a hold-out test set $\mathcal{D}_{test} = (x_i, y_i)_i^N$. We investigate the alignment of the embedding spaces along the curve, i.e. the last layer before the classification layer. We define the absolute embedding space $\mathcal{Z}_k$ as the ensemble of all latent vectors $\{\mathbf{z}_i\}$ in $\mathcal{D}_{test}$ and the space of relative proximity $\mathcal{Q}_k$ as the ensemble of all transformed vectors $\{\mathbf{q}_i\}$ given a set of anchors $\mathcal{A}$ and a similarity function $sim : \mathbb{R}^D \times \mathbb{R}^D \rightarrow \mathbb{R}$. The number of anchors is chosen to match the number of dimensions $D$ of the absolute embeddings. For the VGG16, $D = 512$, and for the ResNet110, $D = 256$. Anchors are sampled randomly from $\mathcal{D}_{test}$, following the procedure of [7]. Table 1 provides an overview of the proposed similarity functions. For the cosine and basis transformations, we center the embedding space $\mathcal{Z}_k$ before calculating similarities using an estimate of the mean based on $\mathcal{D}_{train}$. For the Euclidean transformation, the space is additionally scaled to unit variance. Finally, using the proposed metric, we measure the cumulative alignment for increasing $k$ of both the absolute and relative

embedding spaces. We compare the observed alignment with the alignment of eleven independently trained networks. Code is available here[1]

# 3   Results

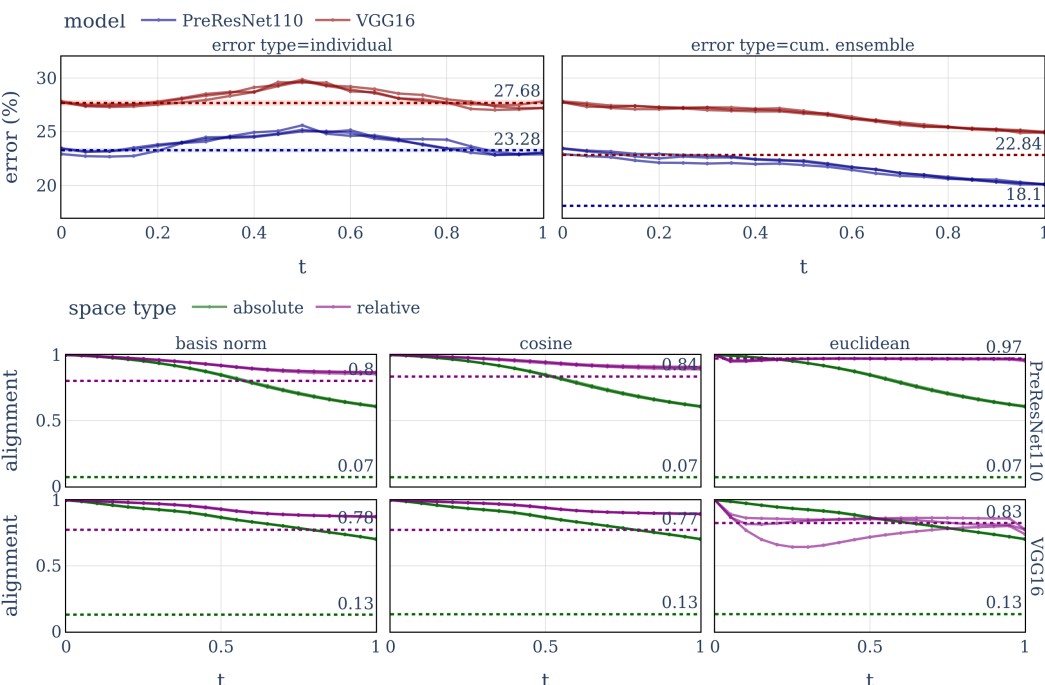

Figure 2: *Top*: Curve fitting results (solid) for VGG (red) and PreResNet (blue), pairwise connecting three modes with individual (left) and cumulative ensemble error rates (right). The individual error rate is the classification error rate for CIFAR100 for a given set of parameters $t$ on the curve. The cumulative error rate is determined by the arg max of the sum of probabilities across all sampled parameters on the curve in $[0, t]$. *Bottom*: Alignment of embedding spaces along the curve for the absolute (green) and relative (purple) space. Dashed lines represent reference values, given as the average error and alignment of eleven independently trained networks.

In Figure 2 we show the individual and cumulative ensemble error rates for each model type and curve. For all models and curves, the test accuracy increases when approaching the midpoint. The cumulative ensemble error decreases with increasing ensemble size. Notably, there is little variation across the pairwise curves. The embeddings of relative representations show higher alignment than the embeddings of absolute representations. This shows that the embedding spaces are confounded by reparametrization. However, the results vary across projection methods. For the VGG, the cosine similarity performs best, whereas for the ResNet it is Euclidean distance. Alignment measured in absolute space decreases almost linearly. The alignment measured in relative space, however, seems to converge. Compared to the baseline experiment, where alignment is measured across eleven independently trained networks, the measured alignments in relative space are closer than those measured in absolute space. Furthermore, the figure shows that the highest negative slope occurs around $t = 0.5$, whereas the slope has limited variation around the endpoints. This finding suggests that ensembles used for uncertainty quantification should be samples far from the endpoints of a connecting curve.

---

[1]https://github.com/fmager/it-s-all-relative

## 4 Discussion

In this work, we investigate the alignment of latent observations when sampling along a curve connecting two modes. Our results show that transforming the spaces into a space of relative proximity reduces uncertainty significantly. This suggests that the uncertainty of ensembles is indeed confounded by reparametrization. The measured uncertainty varies across different relative projections within each modality. This is a limitation, as no general recommendation for a projection method can be made. Each projection will make the latent space invariant to some transformation, and the optimal choice of projection is model-dependent. Certain projections might bias uncertainty estimates. For example, the unit-norm scaling of the cosine similarity measure will increase uncertainties close to the zero vector. In this work, we used a random set of anchors, which eases the workflow of sampling from the curve. However the choice of anchors, e.g. archetypes or centroids, will likely influence the projection quality. The influence of projection function and anchor choice should be further investigated.

## Acknowledgments and Disclosure of Funding

FMM is supported by the Danish Data Science Academy, which is funded by the Novo Nordisk Foundation (NNF21SA0069429) and VILLUM FONDEN (40516) and the Pioneer Centre for AI, DNRF grant number P1. VM is supported by the PNRR MUR project PE0000013-FAIR. LKH is supported by the Pioneer Centre for AI, DNRF grant number P1 and the Novo Nordisk Foundation grant NNF22OC0076907 "Cognitive spaces - Next generation explainability".

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

## A Reparametrization issue and uncertainty in latent space

Consider the linear function $f : \mathbf{x} \to \mathbf{y}$ defined as

$$\mathbf{y} = \mathbf{W}_2 \mathbf{W}_1 \mathbf{x}, \tag{6}$$

where $\mathbf{W}_1 \in \mathbb{R}^{\cdot \times D}$ and $\mathbf{W}_2 \in \mathbb{R}^{D \times \cdot}$ are the learned parameters $\theta$ of the linear map using a loss function $\mathcal{L}(\theta)$. The intermediate representation $\mathbf{z}$ takes the form

$$\mathbf{z} = h(\mathbf{x}) = W_1 \mathbf{x} .$$

It is evident that there exist infinitely many optimal solutions of $\mathbf{W}_1$ and $\mathbf{W}_2$ for the same function $f$. While $f$ remains unchanged, the reparametrization might have a influence on $h$. Consider the following example, where we define $\mathbf{W}_1' = \mathbf{Q}\mathbf{W}_1$ and $\mathbf{W}_2' = \mathbf{Q}^{-1}\mathbf{W}_2$. From Eq. 6 it is easy to see that $f$ does not change, however, the embedding function becomes $h'(\mathbf{x}) = \mathbf{Q}\mathbf{W}_1\mathbf{x} = \mathbf{Q}\mathbf{z}$.

Given two samples of a latent vector $\mathbf{z}_n$ as $\mathbf{z}_n^1 = h(\mathbf{x}_n)$ and $\mathbf{z}_n^2 = h'(\mathbf{x}_n)$, the variance estimate according to Eq. 1 becomes

$$\sigma^2(\mathbf{z}_n) = \frac{1}{2}[(\mathbf{z}_n^1 - \bar{\mathbf{z}}_n)^2 + (\mathbf{z}_n^2 - \bar{\mathbf{z}}_n)^2] \tag{7}$$

$$= \left( \frac{\mathbf{I} + \mathbf{Q}}{2} \, \mathbf{z}_n^1 \right)^2 \tag{8}$$

where $(\mathbf{z}_n^1 - \bar{\mathbf{z}}_n)^2 = (\mathbf{z}_n^2 - \bar{\mathbf{z}}_n)^2$ and $\bar{\mathbf{z}}_n = \frac{\mathbf{I}+\mathbf{Q}}{2}\mathbf{z}_n^1$. Consider now the transformation $\mathcal{T} : \mathbf{z} \to \mathbf{q}$ proposed by [7]. The transformed latent vector $\mathbf{q}$ is expressed relative to a set of anchors $\mathcal{A} = \{\mathbf{z}_j\}_j^A$. Usually one chooses the number of anchors to match the dimensionality of $\mathbf{z}$, in which case one can write $\mathcal{A}$ as a square matrix $\mathbf{A} \in \mathbb{R}^{D \times D}$, where the $d$'th row vector $\mathbf{a}_d$ is the $j$'th latent vector $\mathbf{z}_j$ in $\mathcal{A}$. Using a similarity function $sim : \mathbb{R}^D \times \mathbb{R}^D \to \mathbb{R}$, [7] defines the transformed latent vector $q_i$ as

$$\mathbf{q}_i = [sim(\mathbf{z}_i, \mathbf{a}_1), \ sim(\mathbf{z}_i, \mathbf{a}_2), \ ..., \ sim(\mathbf{z}_i, \mathbf{a}_D)] . \tag{9}$$

If we use the cosine similarity as a similarity function and pose the same reparametrization problem to the transformed latent vector $q_i$, the $d$'th element of the transformed latent vector $\mathbf{q}_i' \in \mathbb{R}^D$ is calculated as

$$q_{id}' = \frac{(\mathbf{Q}\mathbf{z}_i)^T \mathbf{Q}\mathbf{a}_d}{||\mathbf{Q}\mathbf{z}_i|| \, ||\mathbf{Q}\mathbf{a}_d||} = \frac{\mathbf{z}_i^T \mathbf{Q}^T \mathbf{Q}\mathbf{a}_d}{||\mathbf{Q}\mathbf{z}_i|| \, ||\mathbf{Q}\mathbf{a}_d||} \tag{10}$$

Equation 10 shows that Relative Representation with a cosine similarity makes the latent space invariant if $q_{id}' = q_{id}$. This holds true for rotation and scaling transformations, i.e $\bar{\mathbf{Q}} = \alpha\mathbf{U}$, where $\mathbf{U}^T\mathbf{U} = \mathbf{I}$.

## B Latent space sampling

The following is a reformulation of the curve finding experiments by [1]. Consider two fully converged models $f_1$ and $f_2$ with parameters $\hat{w}_1$ and $\hat{w}_2$. A Bezier curve $\phi_{12}(t)$ with the endpoints fixed at $\hat{w}_1$ and $\hat{w}_2$ and a single bend is defined as

$$\phi_{12}(t) = (1-t)^2 \hat{w}_1 + 2t(1-t)\theta + t^2 \hat{w}_2, \quad 0 < t < 1 . \tag{11}$$

Here, $\theta$ are the parameters of the curve and $t$ is the interpolation variable along the curve, where $\phi_{12}(t=0) = \hat{w}_1$ and $\phi_{12}(t=0) = \hat{w}_2$.

Note that the number of parameters of $\theta$ is equivalent to the number of parameters in $\hat{w}_1$ or $\hat{w}_2$ times the number of bends along the curve. The optimal curve is found by minimizing the loss $\mathcal{L}$ below the entire curve, which is defined as

$$l(\theta) = \int_0^1 \mathcal{L}(\phi(t)), dt = \mathbb{E}_{t \sim U(0,1)}[L(\phi_\theta(t))] ,$$

where $\mathcal{L}$ is the loss function used to find $\hat{w}_1$ and $\hat{w}_2$.

## C Transformations

Table 1 shows transformations of a latent obersvation $\mathbf{z}_n$ into a latent observation of relative proximity $\mathbf{q}_n$ based on a set of anchors $\mathcal{A} = \{\mathbf{z}_i\}_i^A$.

| Transformation | Description | |
|---|---|---|
| Rel. Cosine | Basis Transformation, such that the new basis is the cosine distance of each anchor to each point | $\mathbf{q}_n = \frac{\mathbf{z}_n \mathbf{A}^T}{\|\mathbf{z}_n\|\,\|\mathbf{A}_i\|}$ |
| Rel. Euclidean | Basis transformation, such that the new basis is the euclidean distance of each anchor to each point | $\mathbf{q}_n = [q_1, q_2, ..., q_D]$, where $q_i = \|\mathbf{z}_n - \mathbf{a}_i\|_2$ for $i \in [1, ..., D]$ |
| Rel. Basis | Change of basis based on anchor points | $\mathbf{q}_n = \frac{\mathbf{z}_n \mathbf{A}^T}{\|\mathbf{A}_i\|^2}$ |

Table 1: Summary of transformations for a latent vector $\mathbf{z}_i \in \mathbb{R}^D$ and a matrix of anchors $\mathbf{A} \in \mathbb{R}^{D \times D}$, where each row vector $\mathbf{a}_i$ in $\mathbf{A}$ corresponds to an element in the set of anchors $\mathcal{A} = \{\mathbf{z}_i\}_i^A$

.

