# OpenReview forum: "It's All Relative: Relative Uncertainty in Latent Spaces using Relative Representations"
_NeurIPS.cc/2024/Workshop/UniReps — UniReps_

### Official Review · Reviewer_f7F9 · 2024-10-04
**Mathematically sound but lacking in clarity, while the model choices and the weak significance of the results limit its impact and generalizability**

**Rating:** 6
**Confidence:** 4

**Review:**

Most approaches are mathematically reasonable and valid, while lacking clarification details probably due to the page limit. However, further illustration could always fit into the appendix, such as the selection of anchors, the exact definition of relative embedding space, the transformation matrix and how it influences whether the reparameterization introduces additional variance, why scaling to unit variance is appropriate for Euclidean distances in the context of embedding space. Here are more details regarding this issue:

- The Appx. A contains the explanation of the reparametrization and the linear transformation matrix Q applied to the weights, latent vectors, and even latent space. mathematically reasonable, though some middle process should be illustrated, such as the definition of Q should be addressed when it first appears.
- However, Appx. A does not provide evidence to the statement in your last part of introduction “Due to the reparametrization however, the latent uncertainty σ2(zi) will be overestimate”.  which means that your transformation Q change the latent spread of the hidden representation, while orthogonal rotation and uniform scalar won’t introduce additional variance in most cases (except close to the zero vector). Under the same condition, the other statement hold true: “In Appx. A we show that, for certain transformations, the relative representation framework can eliminate σR2 (zi)”.
- If the author mean that the transform matrix Q can be anything, which probably would introduce additional uncertainty during reparameterization, while they could make Q=αU, where U^T U=I which won't lead to overestimation of uncertainty in most cases, then they need to be specifically illustrate these logics. If the author meant that Q=αU always holds, then the I question whether transformation Q would lead to more uncertainty in reparameterization.
- the equation 2:  σO2=σR2+σM2  assume uncertainty (variance) is additive, which might hold when 2 components are independent. But it might not always hold in practice. If reparametrization introduces transformations that interact with the model's uncertainty, the relationship could be more complex than direct addition. In many cases, uncertainties combine in a more nuanced way, such as through variance propagation (e.g., σO2=σR2+σM2+2Cov(σR,σM)).


The results and conclusions are relatively weak, so does the significance of the study. it’s a validation of the method for training ensemble of models developed in the previous literature. It’s great that transform of the hidden layers of the ensemble models increased their alignment, but it is not surprising as long as good transformation matrix are specified. The conclusion about similarity metric selection is model-specific, while only 2 relatively older models are used: VGG-16 (developed in 2014) and Preactivation-ResNet-110 (developed in 2016). More recent architectures such as Vision Transformers (ViTs) or newer variations of efficient convolutional networks could be used (e.g., EfficientNet, ConvNeXt), whereas the VGG and RestNet could be baselines. Limitations of the architecture of the models used here could be stressed or discussed but not. There is also no explanation of the continued relevance of older architectures to the current question. Additionally, I question if the current paper offer insights that can be generalized to more complex tasks, such as multi-task learning, domain adaptation, or few-shot learning. The uncertainty varies across different relative projections for different modality, and even model-specific.

---

> ### Author Response · Authors · 2024-11-07
> **Review Reply**
>
> Dear reviewer,
> Thank you very much for taking the time to read our work. We appreciate your thorough and constructive comments.
> - We agree that a more recent architecture would strengthen our findings, and we plan to add additional models in future work.
> - More details have been added in Appendix A.
> - Wrt. eq. 2, we agree that there could be an interaction of model and reparametrization uncertainty. However, we believe independence is a valid assumption as the initialization is independent of the data.
> - We agree that the work needs further research to understand the transformation posed by reparametrization and the inductive biases introduced by relative transformations with certain distance metrics.

---

### Official Review · Reviewer_nZ2H · 2024-10-06
**Interesting work, but clarity needs improvement**

**Rating:** 6
**Confidence:** 3

**Review:**

This paper proposes using relative representations for an ensemble of embedding spaces that are interpolated along a curve connecting two independently trained NNs with different initializations. Using relative representations, as opposed to absolute representations, enables the model uncertainty of embeddings/representations to be captured more reliably by reducing the reparametrization uncertainty.

**Pros:**
* This work provides interesting insights for how to sample ensembles along a connecting curve: sample far away from the endpoints.
* The experimental result shows the benefits of using relative representations over absolute representations in reducing reparametrization uncertainty (Figure 1 Right).
* A good discussion of the limitations is provided in the discussion section.

**Cons:**
1. Lines `60-61`: The authors mention that "the relative representation framework can eliminate $\sigma_R^2(z_i)$". However, I couldn't find a discussion of this in the main paper and Appendix A. Why and how does the relative representation framework eliminate $\sigma_R^2(z_i)$, i.e., the uncertainty caused by reparametrization?
2. The clarity and presentation requires improvement. For example, the motivation of the paper could be discussed more clearly in the introduction. Clearly discussing important problems such as the item (1) mentioned above would also help readers gain a better understanding.

**Questions:**
* Figure 2: How are the error rates measured? Are relative or absolute representations used for computing the error rates for the models?

---

> ### Author Response · Authors · 2024-11-07
> **Review Reply**
>
> Dear reviewer,
> Thank you for taking the time to read our paper. We would like to clarify our argument regarding $\sigma_R^2(\mathbf{z})$. In Appendix A we show that the Relative Representation framework with a cosine distance metric can eliminate rotation and scaling transformations. Thus, if scaling and rotations are part of transformations caused by reparametrization on the latent space, the framework can remove those transformations and thereby eliminate their impact on uncertainty estimation. We have added a few lines to clarify this argument.

---

### Official Review · Reviewer_evH1 · 2024-10-06
**Revision of UniReps '24 — #39**

**Rating:** 5
**Confidence:** 4

**Review:**

**Paper summary**

In this paper, the authors explore uncertainty in neural networks' latent spaces by focusing on reparameterization issues that distort uncertainty estimation, proposing to address these problems using relative representations, which compare latent space features relative to fixed anchor points. Specifically, they define an alignment score $\rho{}$, which estimates the signal-to-noise ratio for a given ensemble of latent observations. The alignment score $\rho{}$ is a measure of separability between N latent points given $K$ samples each, similar to Fisher’s Linear Discriminant objective for $N$ classes. Then, the authors show empirically that when sampling models along the curve $\phi{}$, transforming the embeddings into a space of relative proximity increases alignment between latent observations. Finally, they also show that the latent observations $\phi{}(t)$ for the curve have high alignment around the endpoints and little alignment at the bending point, suggesting limited information gain when sampling around the endpoints and high information gain when sampling across the midpoint.

**Paper strengths**

The intuition behind our alignment metric is the following is interesting and addresses a significant challenge in understanding latent spaces.
Furthermore, the authors clearly define the problem and provide detailed mathematical formulations to support their proposal (as far as possible given the 4 pages available), and by demonstrating that most latent space uncertainty arises in the middle of the interpolation curve between networks, the paper provides practical insights for building neural network ensembles.

**Paper weaknesses**

The results show that transforming the spaces into a space of relative proximity reduces uncertainty significantly. This finding suggests that the uncertainty of ensembles is indeed confounded by reparametrization. However, the measured uncertainty varies across different relative projections within each modality, and the anchor choice will likely influence the projection quality. As a result, further in-depth research is necessary to evaluate the proposal thoroughly.

---

> ### Author Response · Authors · 2024-11-07
> **Review Reply**
>
> Dear reviewer,
> Thank you very much for taking the time to read our work. We agree that the work needs further research to understand the transformation posed by reparametrization. As we also mention in the discussion of our work, a limitation of the Relative Representation framework used in this context is the inductive bias introduced by a given distance metric, which will lead to different uncertainty estimates.

---

### Official Review · Reviewer_f3uu · 2024-10-07
**Empirical analysis of uncertainty in latent spaces with the relative representations.**

**Rating:** 6
**Confidence:** 3

**Review:**

Strengths:

Originality: A new alignment score is defined based on previous work, which was cited quite well.
Quality: It is a nice paper investigating the uncertainty in the latent space, and the alignment between the latent spaces of same models trained with different initial parametrization. It is a work in progress, and the claims are well-supported. Limitations of the work are addressed too.
Clarity: The paper is mostly well written and organized, with small typos.
Significance: The authors indeed introduce a new approach to measure the alignment between latent spaces and demonstrates the impact of relative representation. It shows preliminary but positive results for further exploration.

Weaknesses / Suggestions:

*  It would be interesting to explore the impact of using different numbers of models (K), on line 89.
* Impact of the number of anchors could be investigated for the future work.
* More comprehensive details on the eleven models used for benchmarking could improve transparency and contribute to reproducibility.
* The rationale behind Eqs. (3–5) is not fully explained. While the reasoning for Eq. (1) and (2) is clear, Eqs. (3–5) need further clarification.
* GitHub link is missing, but probably due to anonymity. Needs to be updated for the final submission.


Minor Edits:
* On line 49, loss is defined using L instead of $\mathcal{L}$
* In Eq. (1), $\mathbf{x}_n$ needs to be replaced by $\mathbf{x}_i$.
* It appears that "N" represents the number of embeddings, though this is not explicitly stated in the paper.

---

> ### Author Response · Authors · 2024-11-07
> **Review Reply**
>
> Dear reviewer,
> Thank you very much for taking the time to read our work. We appreciate your suggestions and will incorporate them as we continue our work, e.g. investigating the number of anchors.
> - All minor edits were corrected
> - The link to the codebase was added
> - We added additional explanations to eq. 3-5
> - As for the number of sampled models K, we find that the change of alignment along the curve is very smooth, so adding more samples does not change the results.

---

### Decision · Program_Chairs · 2024-10-10

**Decision:**

Accept

**Comment:**

In light of the positive reviewers' feedback and relevancy of the submission, we are pleased to accept this paper for presentation at UniReps 2024. We kindly ask the authors to incorporate the reviewers' suggestions and feedback in the final camera-ready version of the manuscript.